# Implementation and adherence of routine pertussis vaccination (DTP) in a low-resource urban birth cohort

Christian E Gunning ,[1] Lawrence Mwananyanda,[2] William B MacLeod,[3] Magdalene Mwale,[2] Donald M Thea,[3] Rachel C Pieciak,[3] Pejman Rohani,[1] Christopher J Gill[3]

[1]Odum School of Ecology, University of Georgia, Athens, Georgia
[2]Right To Care Zambia, Lusaka, Zambia
[3]Department of Global Health, Boston University School of Public Health, Boston, Massachusetts, USA

**Correspondence to**
Dr Christian E Gunning; research@x14n.org

## ABSTRACT

**Introduction** Reliable information on rates of up-to-date coverage and timely administration of routine childhood immunisations are critical for guiding public health efforts worldwide, yet prospective observation of vaccination programmes within individual communities is rare. Here, we provide a longitudinal analysis of the directly observed administration of a three-dose primary vaccination series to infants in a low-resource community in Lusaka, Zambia.

**Methods** Throughout 2015, we recruited a longitudinal birth cohort of mother/infant pairs (initial enrolment, 1981 pairs; attending, 1497 pairs) from the periurban informal settlement of Chawama compound, located in Lusaka, Zambia. We prospectively monitored the administration of scheduled diphtheria–tetanus–pertussis (DTP) vaccinations across the first 14–18 weeks of life. We analysed study attendance and vaccine coverage, both overall and stratified by age group. We employed Kaplan-Meier analyses to estimate delays in age-appropriate administration of vaccine doses. We also assessed schedule timing violations, including early and compressed dose administration.

**Results** At study completion, first dose (DTP1) rates were high (92.9% of attending), whereas third dose completion (DTP3) rates were far lower (61.9%). Missed vaccinations and study dropout both contributed to the low DTP3 completion rates. DTP1 was administered very late (at or after 10 weeks) to 61 infants (4.1%). DTP1 was administered too early to 64 infants (4.3%), and 77 (5.1%) received consecutive doses below the minimum recommended spacing of 28 days.

**Conclusions** We observe substantial individual variation in the timing of early childhood DTP doses, though following this birth cohort proved challenging. Our results indicate that timely administration of both DTP1 and DTP3 remains a challenge in this community. These directly-observed, individual-based results provide an important counterpoint to more course-grained, survey-based national and province estimates of up-to-date vaccine coverage. This study also highlights the challenges of vaccine hesitancy and suboptimal utilisation of (no-cost) healthcare services in a low-resource urban setting.

## Strengths and limitations of this study

► To our knowledge, this is the first prospective urban birth cohort study in a low-resource setting to directly document vaccinations at the place and time of administration.
► By following individuals across time, we are better able to assess healthcare utilisation behaviour relative to cross-sectional surveys.
► This study provides a detailed view of a single community across a year, and may not capture long-term trends or generalise to other populations.

from identifying at-risk communities and age groups for supplementary immunisation activities[1 2] to their use in disease transmission models and subsequent inference.[3–5] Coverage estimates are also key indicators of national vaccination implementation success.[6] As such, high-quality vaccine coverage estimates are critical in shaping modern public health policy.

Estimates of childhood *up-to-date* vaccine coverage from cross-sectional studies (eg, Demographic and Health Surveys (DHS)[7] and UNICEF[8]) have revealed numerous factors affecting vaccine administration in low-income and middle-income countries (LMICs).[9–13] However, these coarse-grained estimates can mask complexities in compliance and implementation. Particularly in LMICs, spatial and socioeconomic heterogeneities in coverage are common, as are delays in the timely administration of vaccine doses.[9 14] Further, the reliability of surveys based on personal and informal records, that is, vaccine cards and maternal recall, remains unclear.[15]

With few exceptions,[16 17] detailed observations of childhood vaccinations during routine clinical care are lacking. Granular, community-based observations of individual patient histories can reveal temporal

## INTRODUCTION

Estimates of vaccine coverage play a central role in epidemiology and public health,

patterns in schedule adherence, including delays in age-appropriate vaccinations. Prospective observation of a cohort can also reveal if community-level diphtheria–tetanus–pertussis (DTP) coverage differs substantially from regional coverage, and can help identify behavioural factors that affect the success of vaccine implementation programmes, including vaccine hesitancy.

Here, we provide a detailed analysis of infant vaccination in a sub-Saharan low-resource urban setting in Lusaka, Zambia. Throughout 2015, we followed a longitudinal birth cohort of mother–infant pairs from birth through 14 weeks of age as part of a broader study of pertussis transmission. Participants were scheduled for six clinic visits total; the primary three-dose DTP series was administered at three of these visits according to the Zambian national schedule of 6, 10 and 14 weeks of age. Our team prospectively documented all clinic visits and administration of DTP doses using an electronic data capture system.[18] We quantified the DTP series dose trajectory of individual infants, and assessed the proportion of infants with up-to-date vaccinations, or who missed one or more doses. We evaluated compliance with published vaccine schedule guidelines, including minimum age restrictions and dose spacings.

In short, our use of directly observed age-specific DTP administration data in a typical LMIC community clinic offers a unique window into vaccine implementation at the individual patient level that can help inform vaccine policy decisions globally.

## METHODS

### Study design

The Southern Africa Mother Infant Pertussis Study (SAMIPS) was a longitudinal birth cohort study conducted in Lusaka, Zambia from March 2015–February 2016 to assess the incidence of pertussis in a representative population of otherwise healthy infants (details provided elsewhere[18]). Mother–infant pairs were enrolled from the population of the Chawama compound, a densely populated periurban slum of approximately 150 000 residents near central Lusaka. The study goal was to capture all live births that occurred in Chawama compound during the study period and, prior to study initiation, a public outreach campaign provided pregnant Chawama residents with information about the study. Infants were enrolled at the Chawama Primary Health Clinic (PHC) during their first scheduled postpartum well-child visit (at approximately 1 week old). The PHC is the only government-supported clinic in this community and, as such, is the primary source of medical care for Chawama residents. This feature allowed us to maximise study reach by capturing the majority of participant healthcare consumption in this community during the study period from a single point of service.

Enrolment eligibility required that infants were born after 37 weeks, weighed over 2500 g, and were delivered without complications or apparent disease. Eligibility also required signed consent, Chawama residency (anticipated remaining in the community during study period), that the HIV status of mothers was known prior to delivery, and that HIV+ mothers were receiving prophylactic antiretroviral therapy at the time of delivery.[18] An extended discussion of study size is provided in.[18]

Mothers were incentivised to join and remain in the cohort in three ways. First, the SAMIPS medical staff provided all routine and acute medical care for study participants during their time of enrolment. This significantly reduced clinic waiting times for care from over 3 hours to half an hour or less. Second, mothers received a travel stipend for each visit valued at approximately US$7. Lastly, a small gift of baby supplies was provided for mothers attending the final scheduled study visit.

Enrolled infants were scheduled for six routine clinic visits at 2–3 week intervals through 14 weeks old (maximum 18 weeks). At each clinic visit, nasopharyngeal swabs were obtained from both mother and infant, and possible pertussis symptoms were recorded. Regular clinic staff provided routine childhood vaccinations at no cost. Administration of DTP doses 1–3 was scheduled at visits corresponding to 6, 10 and 14 weeks of age according to Zambian national guidelines (see WHO Expanded Programme of Immunization (EPI) schedule[19]). DTP vaccinations were administered as a pentavalent combination (Pentavac, Serum Institute of India Limited, Pune, India) that included whole-cell pertussis, *Haemophilus influenzae* type B, and Hepatitis B. The pneumococcal conjugate vaccine was coadministered with DTP; oral rotavirus vaccine administration was scheduled for 6 and 10 weeks of age.

At each visit, infant vaccination cards ('under 5 cards') were reviewed by clinic staff, and the dates of any previously unrecorded vaccinations were noted. When participants missed a scheduled visit, clinic staff attempted to contact participants by phone to reschedule the visit.

### Patient and public involvement

This research was done without patient involvement. Patients were not invited to comment on the study design and were not consulted to develop patient relevant outcomes or interpret the results. Patients were not invited to contribute to the writing or editing of this document for readability or accuracy.

### Assessment of vaccine administration

For protection against pertussis, the WHO recommends that infants receive a three-dose primary series with a DTP-containing vaccine.[19] Guidelines specify a minimum administration age of 6 weeks for the first dose (DTP1), a minimum interval of 4 weeks between subsequent doses and a target interdose interval of 4–8 weeks. In Zambia, pregnant mothers receive one or more doses of tetanus toxoid monovalent vaccine during pregnancy.

For this analysis, we use the Zambian 6–10–14 week EPI schedule for DTP to define target age windows in which infants should have received each DTP dose: window 1,

41–69 days; window 2, 70–97 days; window 3, 98–126 days. Note that window 1 extends 1 day early to account for uncertainty the exact timing birth and vaccination. We tabulate the number of doses each infant received in each window, and at study completion. For example, an infant might receive no doses during window 1, and a dose in both windows 2 and 3, for a total of two doses at study completion.

Infants who received all scheduled doses to-date were categorised as up-to-date. However, once a window was missed, an infant was no longer up-to-date during subsequent age windows. Infants who received no doses by the end of the dose window were categorised as unvaccinated. Vaccinations that occurred >1 day prior to window 1 were classified as early. Vaccinations administered less than 4 weeks after a preceding dose were classified as compressed.

## Data and statistical analyses

Administration of all DTP doses by clinic staff during the study period was directly observed and documented by study staff. All data were manually recorded on standardised data sheets, and digitised using a digital pen electronic data capture system (XCallibre, Durban, South Africa, https://www.xcallibre.com). Data were deidentified prior to analysis: each mother–infant pair was assigned a random date offset of ±3 days, and all study dates (ie, birth and visits) were adjusted by this offset.

Mother–infant pairs were considered attending (on-study) from enrolment until their last recorded clinic visit during the study period. We exclude from further consideration individuals who enrolled in SAMIPS but did not attend any scheduled clinic visits.

We report the number of infants who received exactly (and at least) 0–3 doses by study completion, along with the percent of attending study infants in each of these groups.

We stratified dose outcomes by age window. For each window, we report the number of *unique* infants attending at least one scheduled visit. We also report the number of infants who: received or missed a vaccine dose, received compressed or early vaccinations, and were up-to-date (along with respective percentages). Since attrition and irregular attendance are expected in cohort studies, we compute percentages relative to the number of unique infants in each window (above).

We perform a modified Kaplan-Meier analysis of vaccination delays.[14 20] The standard Kaplan-Meier method assumes that non-attending individuals (ie, right-censored) experience events at the same rate as attending individuals. Here we explore an alternative worst-case scenario where non-attending individuals receive no further vaccinations during the study period (subsequent to study drop-out). While subsequent make-up vaccinations may occur, our assumption of minimal off-study vaccination is plausible, as the Chawama clinic is the sole provider of DTP in this community, and study staff recorded all DTP doses there during the study period.

## RESULTS

### Participation

Infants were enrolled in the study between March and December 2015 (1981 out of 3033 screened). The five most common reasons for study exclusion include lack of consent (14.3%), low infant birth weight (7.9%), community non-residence (5.9%), mother anticipates moving (4.1%) and mother not 18–39 years old (3.7%) (see Supplemental Table 3 in[18] for enrolment details). Enrollees that did not attend any subsequent clinic visits (484 total) were removed from this analysis, leaving 1497 initially attending infants. We depict study participation over time of the remaining infants in figure 1A, while figure 1B,C details the timing of study events. At the beginning of windows 1–3, 1400 infants (window 1), 1306 infants (window 2) and 1089 infants (window 3) were still attending (see online supplemental figure 1 for study profile). The observed drop-off in cohort size near the end of window 3 represents graduation from the study, as infants' final clinic visits (and DTP dose) were scheduled for approximately 14 weeks of age. Overall, the duration of study attendance was a median of 96 days (min=12 days, max=120 days, excluding one recorded vaccination at age 272 days).

### Demographic characteristics

Overall, the demographic characteristics of participants did not covary with duration of study attendance (see table 1), including all enrolled individuals, individuals attending at least one scheduled visits (attending), and individuals attending all scheduled visits (completed). Mothers' median age was 25 years, regardless of attendance. Across enrolled mothers, 17.5% were HIV+, while slightly more attending mothers were HIV+ (19.3%). Infants' median birth weight was 3000 g. Regardless of attendance, median household size was 1 infant of <1 year of age (including study infant) and 2 children of <5 years of age (including infants).

With one exception, demographic characteristics did not affect the relative risk of the following end-of-study outcomes: being unvaccinated, receiving at least one dose (DTP1) and receiving all three doses (DTP3) (figure 2). We did observe a marginally lower risk of DTP3 (ie, higher risk of incomplete vaccination) in infants who received oral polio vaccine (OPV) at birth (RR=0.89, 95% CI 0.81 to 0.97). However, we observed no consistent direction of associations, and we note that the observed associations are consistent with sampling variation arising in multiple comparisons.

### Scheduled doses

Vaccination outcomes at the completion of the study (total doses received) are summarised in table 2, while table 3 details the cumulative doses received at the end of each age window. Overall, approximately 7.1% (107/1497) of attending infants had not received any DTP vaccinations at the conclusion of the study (table 2). This corresponds to 9.8% (107/1089) of infants who remained in the

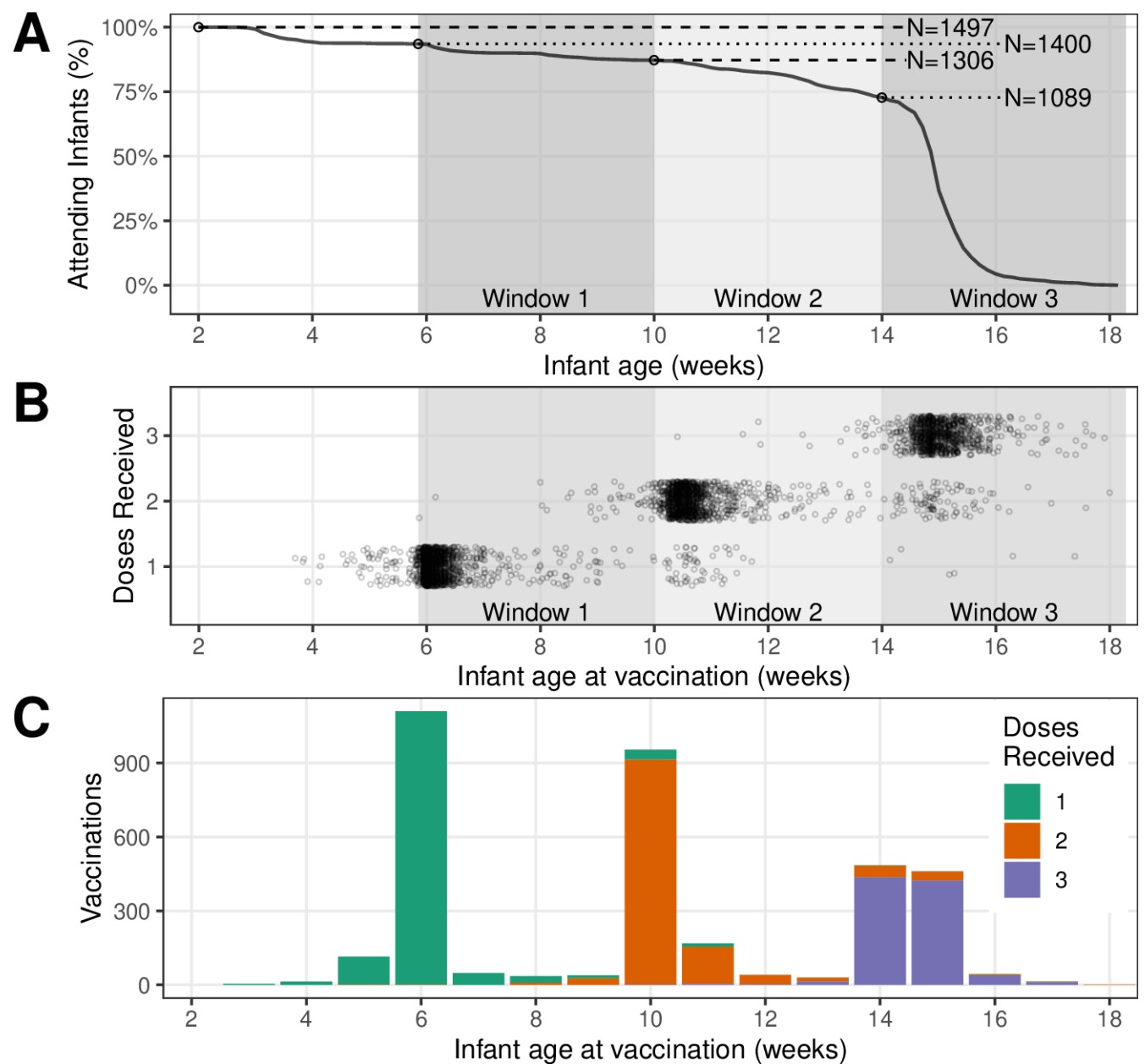

**Figure 1** (A) Study participation by infant age. Shaded regions show scheduled administration of DTP doses 1–3. (B) Timeline of diphtheria–tetanus–pertussis dose administration (see also online supplemental figure 2). (C) Number of vaccinations per week, by dose number.

cohort up to the beginning of window 3 (and thus had the opportunity to receive each of the three scheduled doses, see figure 1A).

The timing of DTP dose administration to individual participants is shown in figure 1B (for a detailed view of all clinic visits, see online supplemental figure 2). We also aggregate these data by week, highlighting that most infants received each DTP dose within a week of the target schedule of 6, 10 and 14 weeks of age (figure 1C). However, deviations from the planned schedule were common. For example, early DTP vaccinations were administered to 64 infants prior to window 1 (table 3), before which no vaccinations should have occurred (except for doses of the oral polio and BCG vaccines

administered at birth, see table 1). Similarly, the spread of events to the right of the target administration age of 6, 10 and 14 weeks represents vaccination after the scheduled delivery date (figure 1B–1C). Two general trends here are apparent across the three scheduled doses. With each subsequent dose: (1) the average time of administration drifts farther to the right of (later than) the target schedule and (2) the clustering of administration times becomes more dispersed.

We also observed infants who missed a dose and later received a subsequent dose on (or near) schedule (figure 1B–C). For example, 118 infants received their first DTP dose during window 2, while 170 infants received a second DTP dose during window 3 (table 3, figure 1C).

**Table 1** Demographic characteristics of participants by duration of study attendance (columns), showing percentage or median, as appropriate

| Duration of parameter | Enrolled | Attending | Completed |
|---|---|---|---|
| Number under study | 1981 | 1497 | 834 |
| Mothers | | | |
| Married | 90.2% | 90.1% | 90.3% |
| HIV+ | 17.5% | 19.3% | 18.6% |
| Median age | 25 (21–29) | 25 (22–29) | 25 (22–30) |
| Median infants in house (<1 year) | 1 (1–1) | 1 (1–1) | 1 (1–1) |
| Median children in house (<5 years) | 2 (1–2) | 2 (1–2) | 2 (1–2) |
| Infants | | | |
| Born at Chawama PHC | 56.9% | 56.7% | 58.2% |
| Born at University Teaching Hospital | 34.8% | 35.5% | 34.3% |
| Female sex | 46.9% | 46.4% | 46.4% |
| Received BCG at birth | 46.4% | 46.6% | 46.2% |
| Received OPV at birth | 33.4% | 32.4% | 27.7% |
| Median birth weight (kg) | 3 (2.8–3.3) | 3 (2.8–3.3) | 3.1 (2.8–3.3) |

IQR is shown in parentheses. Attending mother–infant pairs attended one or more scheduled clinic visit. Completed individuals attended all six scheduled visits.
OPV, oral polio vaccine; PHC, Primary Health Clinic.

Since the study did not follow participants beyond age 16 weeks (with few exceptions), it remains unclear whether (and when) catch-up of any doses missed during the study might have occurred.

Time-stratified results based on infant visits within each age window are shown in table 4. These results are less sensitive to cohort attrition, as they assess outcomes over a shorter time period. Here, we find that up-to-date rates fell across the study from 95.8% of visiting infants (window 1, DTP1) to 88.4% (window 2, DTP2) and finally to 83.7% (window 3, DTP3). The unvaccinated rate of visiting infants also fell from window 1 (12.3%) to window 2 (8.8%), but did not decrease further during window 3 (9.9%). We note that many infants who visited the clinic in windows 1–3 nonetheless missed scheduled vaccinations: 6.6% (window 1), 9.1% (window 2) and 8.4% window 3.

### Vaccine timing
A timely initial dose plays an important role in protecting infants from preventable morbidity and mortality[21]; prior to DTP1, study infants' protection against pertussis

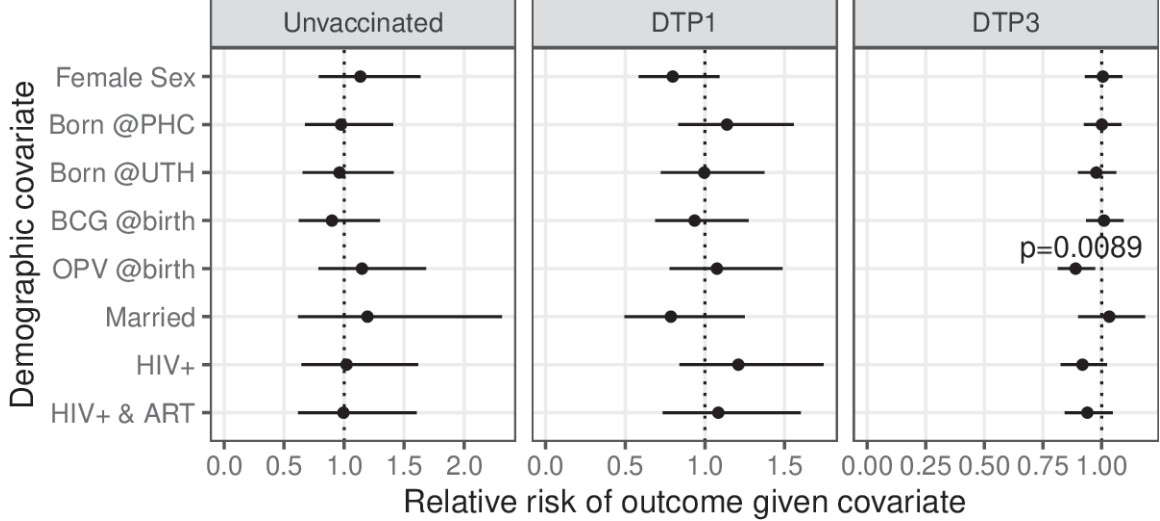

**Figure 2** Relative risk of vaccination outcome (columns) by demographic covariate (y-axis), showing 95% CI. Select p values are shown above estimates. Covariates: @birth indicates the respective vaccine was administered at birth; HIV+ & ART indicate the mother was HIV+ and taking antiretroviral therapy at time of enrolment. See table 2 for outcome frequencies. OPV, oral polio vaccine; PHC, Chawama Primary Health Clinic. UTH, University Teaching Hospital.

**Table 2** Vaccine status at study completion

| N | Exactly N doses (%) | At least N doses (%) |
|---|---|---|
| 0 | 107 (7.1) | 1497 (100.0) |
| 1 | 147 (9.8) | 1390 (92.9) |
| 2 | 316 (21.1) | 1243 (83.0) |
| 3 | 927 (61.9) | 927 (61.9) |

Number of infants that received at least (or exactly) 0–3 doses of diphtheria–tetanus–pertussis at study completion. Percentages are rounded (out of 1497 attending infants).

depended on unknown but presumably low and waning maternally derived immunity. We observed 61 infants (4.1% of attending) that received DTP1 at 10 weeks of age or greater, though only 6 (0.4% of attending) received DTP1 at 14 weeks of age or greater (table 3, figure 3).

Our modified worst-case Kaplan-Meier survival analysis (assuming no vaccination in infants lost to follow-up) revealed a modest median delay in the timely administration of DTP1 (1 day), though the median delay grew across primary series to 5 days (DTP2) and 9 days (DTP3). However, many infants experienced more substantial delays: for DTP1 the upper decile delay (slowest 10%) was 32 days, and for DTP2 the upper quantile delay (slowest 25%) was 17 days. Finally, 570 infants (38.1% of attending) did not receive DTP3 during the study.

By comparison, schedule violations involving early and compressed doses were minor. As noted above, DTP1 was administered too early to 64 infants (4.3% of attending), with most of these doses administered 2–7 days early (at age 35–40 days) (table 3, figure 4A). The median spacing between subsequent doses (31 days) slightly exceeded the minimum recommended spacing of 28 days (figure 4B). A small minority of infants experienced dose compression between doses 1–2 and/or doses 2–3: 77 infants received doses spaced less than 25 days apart, while 48 infants received doses less than 20 days apart.

## DISCUSSION

This study offers a descriptive analysis of prospectively collected, carefully documented observations of DTP

**Table 3** Cumulative doses received across study

| | Early | Window 1 | Window 2 | Window 3 |
|---|---|---|---|---|
| Received | 0–40 | 41–69 | 70–97 | 98–127 |
| DTP1 | 64 | 1264 | 55 | 6 |
| DTP2 | | 36 | 1117 | 90 |
| DTP3 | | | 20 | 907 |
| Any | 64 | 1300 | 1192 | 1003 |

Number of unique infants that received diphtheria–tetanus–pertussis (DTP1–3) by the *end* of each age window (columns, in days). Bottom row shows total doses administered in each window (some infants received more than one dose per window). A single infant received DTP1 at age 272 days (not shown).

vaccinations administered in an urban African public health clinic. Overall, we found that 61.9% of attending infants (927/1497) received the full DTP3 series in a timely manner, corresponding to 85.1% of infants who attended into window 3 (and were thus available for a third dose, 791/1089). This is remarkable given that Chawama's EPI implementation lacks direct client outreach or other supportive interventions. However, we were concerned by higher than anticipated rate of missed (and delayed) doses: at study completion, 9.8% of attending infants had received only one dose, while 7.1% of attending infants lacked records for any DTP vaccinations. In short, our findings suggest that access to (no-cost) healthcare, available to all study infants, was not the sole limiting factor in DTP uptake in Chawama compound.

While not directly comparable with our results, WHO-UNICEF estimated that Zambia's 2015 up-to-date DTP3 coverage was 90%,[22] much higher than our findings of 61.9%. However, our DTP1 rates at study completion (92.9%) were proportionally closer to corresponding WHO-UNICEF estimates (97%) and 2013–2014 Zambia DHS estimates for Lusaka province (97.8%)[23] (though the ceiling effect leaves little range for observable differences here). Overall, our results suggest that delays in timely vaccination warrant further attention in Zambia, as does spatial heterogeneity (eg, across socioeconomic gradients).

The longitudinal nature of this study allowed us to resolve key implementation features that cross-sectional or survey-based approaches cannot resolve, including dose timing. In this cohort, early and compressed vaccinations were relatively rare. However, we observed that administration of scheduled vaccinations was more variable and, on average, less timely with each subsequent dose. This pattern indicates that delays are not random across life, and suggests that delays early in life have downstream effects on the timing of subsequent vaccinations. We also gained insights into how individuals' behaviour affected population-level coverage rates. For example, the small change in unvaccinated infants attending window 2 (114, 8.8%) and window 3 (108, 9.9%) suggests that doses are not skipped randomly by study participants. Rather, a core group of individuals appears to have regularly attend clinic visits while opting out of (no-cost) routine vaccines, despite encouragement by our study team. Indeed, earlier qualitative work in this cohort suggests that vaccine hesitancy plays a non-trivial role here.[24]

Following a birth cohort across almost 5 months of life proved challenging in this setting. We experienced high rates of initial non-participation among eligible infants (484 out of 1981 initial enrollees), as well as ongoing attrition of the cohort throughout the study. Indeed, only 72.7% of attending infants (1089, figure 1) participated in the study up to an age of 14 weeks (98 days), and were thus available to receive all scheduled DTP doses during the study. However, our intensive prospective monitoring of the primary source of healthcare in Chawama leads us to believe that enrollees who missed one or more visits

**Table 4** Vaccinations stratified by age window, showing number of unique infants within each age window

| Age window | Target (N) | Visits | Received (%) | Missed (%) | Compressed (%) | Up-to-date (%) | Unvaccinated (%) |
|---|---|---|---|---|---|---|---|
| 41–69 | 1 | 1378 | 1287 (93.4) | 91 (6.6) | – | 1320 (95.8) | 169 (12.3) |
| 70–97 | 2 | 1301 | 1183 (90.9) | 118 (9.1) | 50 (3.8) | 1150 (88.4) | 114 (8.8) |
| 98–127 | 3 | 1093 | 1001 (91.6) | 92 (8.4) | 47 (4.3) | 915 (83.7) | 108 (9.9) |

Visits: infants attending ≥1 scheduled visit. Received, missed: infants that received (or did not receive) a diphtheria–tetanus–pertussis dose. Compressed: infants vaccinated <4 weeks after previous dose (see also figure 4B). Up-to-date: infants that received the targeted Nth dose in this window. Unvaccinated: infants that received zero total doses by end of window. Percentages are rounded (out of visits).

rarely receive unrecorded vaccinations at the PHC during the study period. In addition, we expect that non-clinic vaccinations of enrollees during this period were rare due to cost and geographical access. On the other hand, study involvement could have introduced an upwards bias on clinic visit behaviour and DTP uptake relative to this population's background, non-study healthcare behaviour.

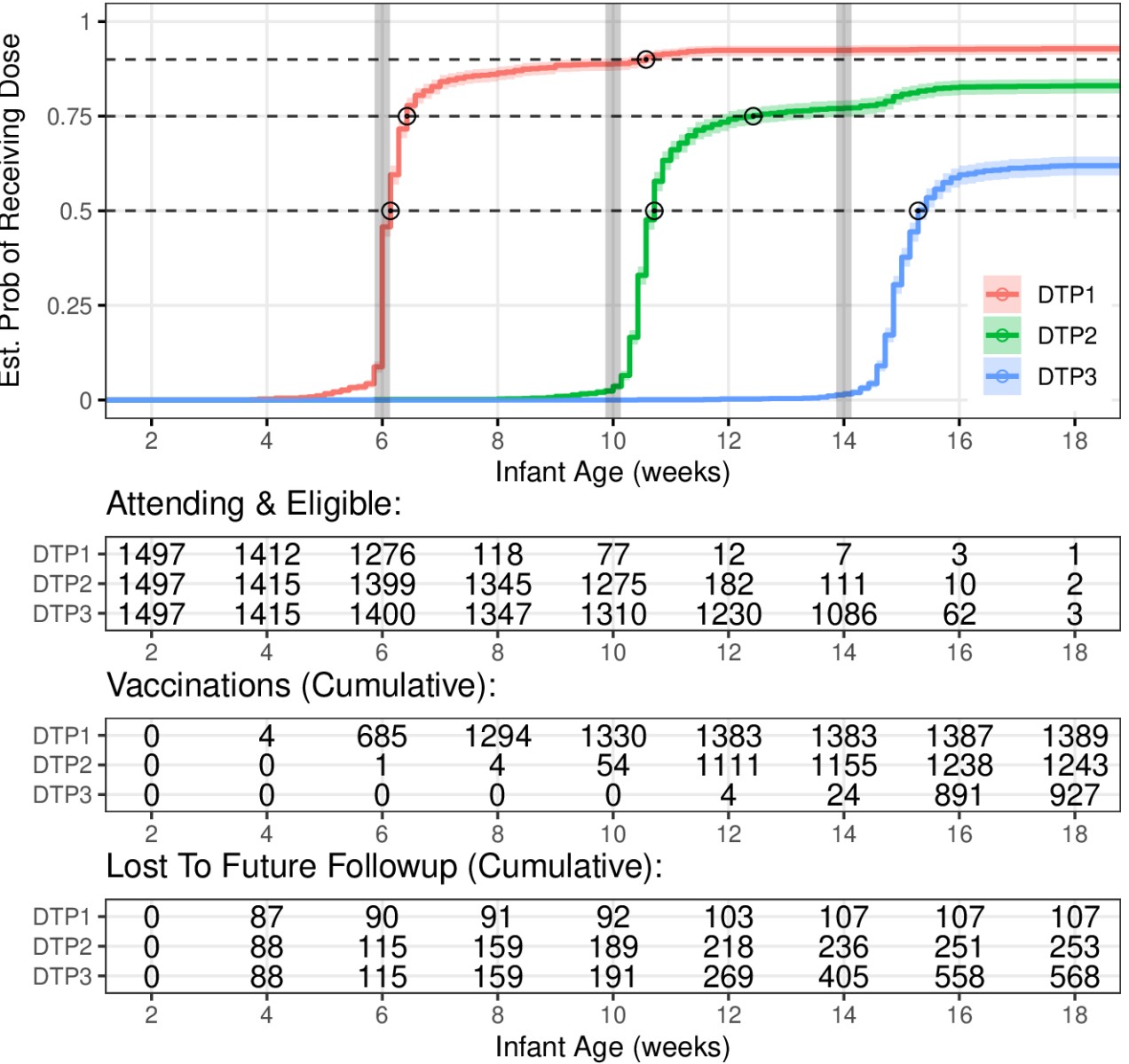

**Figure 3** Modified Kaplan-Meier analysis: estimated dose administration probability for diphtheria–tetanus–pertussis (DTP) doses 1–3, assuming no vaccination in infants lost to follow-up. Vertical grey lines show dose target age; intersections with horizontal dashed lines show the median, upper quantile (slowest 25%) and upper decile (slowest 10%) delays for each dose. Tables show number of infants at select ages.

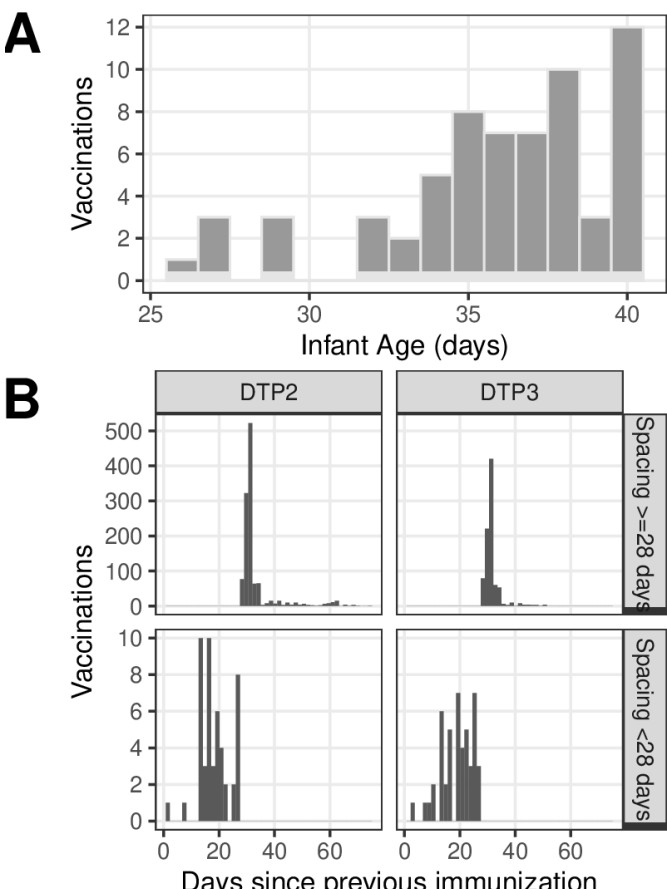

**Figure 4** (A) Number of early vaccinations (diphtheria–tetanus–pertussis (DTP1)) by infant age. (B) Number of vaccinations by dose spacing (time since previous dose), split by dose number (columns). Rows show compressed versus appropriate spacing (rows).

Two unresolved questions remain: (1) whether infants who left the cohort early were subsequently vaccinated at the PHC after study graduation or were vaccinated elsewhere off-study and (2) whether attending infants who missed scheduled doses later received catch-up doses, for example, at subsequent well-child visits. In Lusaka, catch-up of missed doses is available at monthly clinic visits and two targeted 'child health weeks' (June and November). However, these supplemental vaccination campaigns are implemented without respect to known vaccination compliance (as documented in each child's under five card), and thus do not systematically target children who missed doses during the primary series. Moreover, after the final study visit (at age 14–16 weeks), the next routine well child visit is not until 9 months of age, when the measles vaccine dose is given. As such, even when supplemental vaccination occurs, the subsequent delays in timely vaccination may be substantial.

We believe this study fills an important gap in the childhood vaccination literature. As a periurban informal settlement, Chawama represents a typical underserved community in a LMIC, where a single public clinic is the primary healthcare resource for more than 150 000 residents. This study provides an important point of comparison for additional such prospective studies, in order to better assess variation by geographical region or resource availability. We also believe that further investigation of these findings could provide significant value to the public health community. For example, while beyond the scope of this study, a cross-sectional seroconversion survey of young children (eg, 1–5 years of age) could help quantify the frequency of late or absent vaccination stratified by age. For example, seropositivity for tetanus antibodies could provide a robust indicator of (at least) DTP1 uptake, as tetanus toxoid is highly immunogenic and elicits durable immunity.[25] Such surveys could help identify at-risk communities and target additional resources for catch-up campaigns.

Despite lingering uncertainties, our analyses raise broad questions about pertussis control in low-resource settings, including herd immunity and epidemic preparedness. Crowded living situations can facilitate pertussis transmission relative to less dense regions,[26 27] while the scarcity of macrolide treatments in this community may yield longer-lasting infections[28]; both effects could raise the critical vaccination threshold required for herd immunity.[29] Further, lower DTP coverage here relative to Lusaka province is consistent with the previously-described impacts of socioeconomic status.[10–13] Together, these forces could allow dense informal settlements to serve as foci for more widespread pertussis epidemics. Our results also highlight the potential public health consequences of such an epidemic. We directly observed 169 infants (11.3% of attending) who remained unvaccinated at age 10 weeks, along with a corresponding 114 infants (7.6%) at age 14 weeks. These infants were at elevated risk of severe disease, hospitalisation and death,[30 31] yet low-resource settings such as Chawama are ill-prepared to treat severe cases or high case loads. Indeed, the PHC has only 10 hospital beds to serve this community.

These results have additional implications for the full package of vaccines administered via the EPI schedule, as failure to receive a scheduled DTP dose may also indicate failure to receive additionally scheduled vaccinations. For example, both the pneumococcal conjugate and rotavirus vaccines are commonly coadministered with DTP. We expect that marginal improvements in EPI implementation would be highly impactful to public health outcomes, particularly in underserved communities, and that detailed descriptions of when and why EPI schedule violations occur will help identify and remediate existing barriers to vaccine uptake in these communities.

In summary, this study provides a uniquely granular view of routine DTP implementation in a low-resource urban African setting that highlights individual behaviour. Our data suggest that delayed age of first vaccination, along with possible vaccine hesitancy and loss of herd immunity, poses a significant public health risk to this community, echoing calls to strengthen EPI programmes in low-resource urban settings.[32 33]

**Correction notice** This article has been corrected since it first published. The provenance and peer review statement has been included.

**Contributors** Conceived study: CJG. Designed study: CJG and CEG. Conducted analysis, created figures and tables: CEG. Contributed to and interpreted analysis: CEG, WBM, PR and CJG. Wrote manuscript: CEG, PR and CJG. Contributed to manuscript revisions: all authors.

**Funding** The Bill & Melinda Gates Foundation funded the initial SAMIPS cohort (OPP 1105094). The National Institutes of Health/National Institute of Allergy and Infectious Diseases supported all subsequent analyses, including this analysis, using the SAMIPS database (1R01 AI133080).

**Disclaimer** The study sponsors had no role in study design, analysis, or publication.

**Competing interests** None declared.

**Patient consent for publication** Not required.

**Provenance and peer review** Not commissioned; externally peer reviewed.

**Data availability statement** Data are available in a public, open access repository. Extra data can be accessed via the Dryad data repository at http://datadryad.org/ with the doi: 10.5061/dryad.mpg4f4qxs.

**ORCID iD**
Christian E Gunning http://orcid.org/0000-0001-6403-6553

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
