## [Reviewer comments · BMJ Open]

ARTICLE DETAILS

TITLE (PROVISIONAL)	Implementation and adherence of routine pertussis vaccination (DTP) in a low-resource urban birth cohort
AUTHORS	Gunning, Christian; Mwananyanda, Lawrence; MacLeod, William; Mwale, Magdalene; Thea, Donald; Pieciak, Rachel; Rohani, Pejman; Gill, Christopher

VERSION 1 – REVIEW

REVIEWER	Mark McMillan The University of Adelaide, Australia
REVIEW RETURNED	07-Jul-2020

GENERAL COMMENTS	Thank you for the opportunity to review “Implementation and adherence of routine pertussis vaccination (DTP) in a low-resource urban birth cohort”. Vaccine uptake and timeliness is an important issue, especially in low-resource setting. This appears to be an exploratory analysis from participants enrolled in a study investigating the incidence of pertussis among HIV-exposed and unexposed Zambian Infants through 14 weeks of age. Please see my comments and queries below: Introduction: You have conducted this analysis on a study cohort. What isn’t clear in the background is what a routine vaccine program looks like in Lusaka, Zambia. Can you provide a bit more information about that? It also informs the reader about the generalisability of your findings. Are these findings applicable to Lusaka, Zambia when the study is no longer running (for the discussion)? Are they applicable to other parts of Africa with similar programs? Methods: Page 3 Line 25 you may need to reword this sentence, it is currently unclear. Page 4 line 20 should read 41-69 days My main query is if attrition bias impacts on some of the findings you
--

have reported. You have made a case that those who do not return to the clinic are unlikely to be vaccinated *“While subsequent make-up vaccinations may occur, our assumption of minimal off-study vaccination is plausible, as the Chawama clinic is the sole provider of DTP in this community, and study staff recorded all DTP doses there during the study period communities”*

You have lost 24% of your cohort before dose 1. Then those that are lost to follow up are dropped from the 3 vaccination windows, by visit 3 that's close to 45% of your participants. For this type of outcome where you are reporting on uptake and timeliness of vaccination these participants are crucial to your outcome.

You employ Kaplan-Meier analysis that takes into account the participants lost to follow up for timeliness. However, the majority of your reporting is centred around the descriptive data that doesn't take those loss to follow up into account.

I you may want to consider accounting for those lost to follow up in your results as a whole, not just timeliness.

Results:

Page 6 line 40 Is not clear *“(save for doses of the oral polio and BCG vaccines, administered to some infants at birth; see Table 1).”*

Page 6 Line 48 *“Both trends reinforce the general concept that events occurring earlier in life have inevitable down-stream effects on subsequent vaccinations.”* I think this is better in the discussion section. You may need to explain in more detail how you came to that conclusion, it's currently not clear.

Page 7 Line 5 *“Overall, approximately 8.2% of 1,497 attending infants had not received any DTP vaccinations at the conclusion of the study. A similar pattern was observed within each window, where infants who visited the clinic in windows 1-3 nonetheless missed scheduled vaccinations at a rate of 12.2% (window 1), 14.0% (window 2), and 10.4% window 3”*

I think this where the participants lost to follow up impacts on your findings. As your study progresses the participants that missed their scheduled doses becomes less. The reality is that the number of participants that missed their scheduled vaccinations most likely increased in your cohort as a whole. You do address this in your limitations, but I wonder if you are also able to account for this in your results.

The next sentence is the same. *“the unvaccinated rate also fell from 18.5% during window 1 to 11.4% during window 3.”* In your cohort as a whole I don't think that the number of unvaccinated (up to date) participants was likely to be less at window 3.

I realise you are only looking at those participants who returned for their study visits, but these findings are potentially the opposite of what is happening in the Lusaka community, or your complete study

	cohort. The tables have overlapping information. It's worth considering dropping either table 3 or table 4. It's not clear to me why the numbers of infants receiving the vaccine in the windows isn't the same in table 3 and 4. Why is the total of those in window 1 1212 and table 4 it's 1203? Discussion: Page 8 Line 46 "Overall, we found that most infants received the full DTP3 series in a timely manner, indicating successful initiation into the EPI." I think you need to say that this was only in study participants who returned for all 3 study visits.
--	---

REVIEWER	Ane Fisker University of Southern Denmark, Denmark
REVIEW RETURNED	08-Jul-2020

GENERAL COMMENTS	In their paper "Implementation and adherence of routine pertussis vaccination (DTP) in a low-resource urban birth cohort" the authors take advantage of the implementation of a longitudinal study to monitor the pertussis incidence and assesses the DTP (Penta) vaccination coverage and timeliness. While the close follow up certainly provides opportunities for a detailed description of the implementation programme, the extent to which the implementation of the vaccination programme has been affected by the study vaccines will have to be clarified. The link between the study visits and vaccination services is not clear – are these always on the same date? Are the clinic visits scheduled at particular ages? I.e., the timing of the clinic visits every 2-3 weeks, how were these scheduled? From supplementary Fig S2, the density seems high at week 3, 6, 8, 10.5 and 12.5 determined. Results seem to indicate that there are children "catching up" on their timelines, "Infants who missed a dose and later received a subsequent dose on (or near) schedule are also evident in Figures 1B-C. For example, 98 infants received their first DTP dose at age 10 weeks, while 160 infants received a second DTP dose at age 14-15 weeks (Figure 1C)." and P 10, l 16: "For example, the small change in unvaccinated infants attending window 2 (139, 10.7%) and window 3 (123, 11.4%) suggests that doses are not skipped randomly by study participants" I do not follow how this can be inferred from data on the individual vaccination – and not on the individual child. Half of the screened infants enter the study. It is reported that the main reasons for study exclusion were lack of consent (14.3%), low infant birth weight (7.9%), and community non-residence (5.9%) – but these numbers do not come close to the 50% excluded. Has something been forgotten? The numbers in text and figure S1 are not aligned. To make it clear what the reasons for non- participation are, I suggest to add these to Figure S1. "Prospective observation of a cohort can also reveal if community level DTP coverage differs substantially from regional coverage, and can help identify behavioral factors that affect the success of vaccine
---

	implementation programs, including vaccine hesitancy.” – But the authors do not address this. Could the clinical data be helpful in identifying if illness episodes made the mothers defer vaccinations? P 6. 44. “Data were de-identified prior to analysis: each mother-infant pair was assigned a random date offset of +/-3 days, and all study dates were adjusted by this offset.” Presumable “study dates” mean date of birth, date of visit and date of vaccination, but please clarify. The authors conclude their data can be used to identify spatial heterogeneity, but then goes on to compare their estimates with coverage at 12 months Figure 2 – Better labels of the y-axis would make the figure easier to read. Figure 3 – Last tables – I do not understand how 93,92 and 92 DTP1-2-3 children can be lost to FU at 4 weeks. P. 8, line 40: (save for?)
--	--

VERSION 1 – AUTHOR RESPONSE

Reviewer 1:

Introduction:

You have conducted this analysis on a study cohort. What isn't clear in the background is what a routine vaccine program looks like in Lusaka, Zambia. Can you provide a bit more information about that? It also informs the reader about the generalisability of your findings. Are these findings applicable to Lusaka, Zambia when the study is no longer running (for the discussion)? Are they applicable to other parts of Africa with similar programs?

- We have added information about the Zambian national schedule & its relation to this study to the introduction. We have attempted to clearly denote throughout the manuscript that our study was conducted in an under-served community in Lusaka, and that we expect our results to differ from that of more affluent Lusakans. The results here are relevant to understanding vaccination adherence among the larger population of Lusaka's urban poor, who comprised ~90% of the city's population.
- We agree that generalisability and broader relevance are important points. We have expanded on the discussion to better address the specific relevance of this study to low-resource urban settings more broadly. We also believe this point is also addressed in paragraph 7 of the discussion: “Despite lingering uncertainties, our analyses raise broad questions about pertussis control in low-resource settings...”.

Methods:

Page 3 Line 25 you may need to reword this sentence, it is currently unclear.

- Fixed.

Page 4 line 20 should read 41-69 days

- Fixed.

My main query is if attrition bias impacts on some of the findings you have reported. You have made a case that those who do not return to the clinic are unlikely to be vaccinated “While subsequent make-up vaccinations may occur, our assumption of minimal off-study vaccination is plausible, as the Chawama clinic is the sole provider of DTP in this community, and study staff recorded all DTP doses there during the study period communities”

You have lost 24% of your cohort before dose 1. Then those that are lost to follow up are dropped from the 3 vaccination windows, by visit 3 that's close to 45% of your participants. For this type of outcome where you are reporting on uptake and timeliness of vaccination these participants are crucial to your outcome.

- We agree that attrition remains a significant question in this study, and we expand on this question at some length in the discussion. We have attempted to present our analyses in sufficient detail so that readers can draw their own conclusions regarding unobserved events and behaviors.

You employ Kaplan-Meier analysis that takes into account the participants lost to follow up for

timeliness. However, the majority of your reporting is centred around the descriptive data that

doesn't take those loss to follow up into account. You may want to consider accounting for those lost to follow up in your results as a whole, not just timeliness.

- We attempt to provide a balanced approach here. In particular, p4 L55 onwards describes the methods used for Table 4, which is stratified by time within the study. Our goal here is to provide a snapshot of vaccination that is less affected by loss-to-followup, as attrition within each age window is modest relative to total attrition.

Results:

Page 6 line 40 Is not clear "(save for doses of the oral polio and BCG vaccines, administered to some infants at birth; see Table 1)."

- Reworded for clarity.

Page 6 Line 48 "Both trends reinforce the general concept that events occurring earlier in life have inevitable down-stream effects on subsequent vaccinations." I think this is better in the discussion section. You may need to explain in more detail how you came to that conclusion, it's currently not clear.

- We have moved this point to the discussion, expanded on our reasoning, and softened the language.

Page 7 Line 5 "Overall, approximately 8.2% of 1,497 attending infants had not received any DTP vaccinations at the conclusion of the study. A similar pattern was observed within each window, where infants who visited the clinic in windows 1-3 nonetheless missed scheduled vaccinations at a rate of 12.2% (window 1), 14.0% (window 2), and 10.4% window 3"

I think this where the participants lost to follow up impacts on your findings. As your study

progresses the participants that missed their scheduled doses becomes less. The reality is that the number of participants that missed their scheduled vaccinations most likely increased in your cohort as a whole. You do address this in your limitations, but I wonder if you are also able to account for this in your results.

- We agree that considerable ambiguity surrounds the appropriate choice of denominator here. A primary goal of our study is to report sufficient detail for readers to draw their own conclusions. We note that Tables 2-4 describe three complimentary viewpoints of the same data:
 - Table 2, Vaccination status at study completion (denominator is all attending infants)
 - Table 3, Cumulative doses throughout study (unique participants)

○ Table 4, Vaccinations stratified by time (denominator is currently attending infants)
The next sentence is the same. “the unvaccinated rate also fell from 18.5% during window 1 to 11.4% during window 3.” In your cohort as a whole I don’t think that the number of unvaccinated (up to date) participants was likely to be less at window 3.

- These results are time-stratified (i.e., from Table 4). We have substantially revised our results for clarity: we have group together all time-stratified results, and refer to the denominator here as “visiting infants” to better concord with Table 4 and avoid confusion with “initially attending infants”.

I realise you are only looking at those participants who returned for their study visits, but these findings are potentially the opposite of what is happening in the Lusaka community, or your complete study cohort.

- We respectfully disagree with respect to residents of Chawama compound. Please note that we have not attempted to characterize the vaccination behavior of the city of Lusaka; for comparison, we provide others’ estimates of up-to-date vaccination for Zambia as a whole. We have revised the methods to highlight and clarify that the public health clinic is the only local no-cost source of vaccinations available to study participants, that the majority of health care services of Chawama residents is provided by this single clinic, and to detail participant compensation. Given our prospective surveillance of this clinic and provided study compensation, we expect that we successfully captured most clinic interactions during the study, and that (for-cost) non-clinic vaccinations of any study infants were rare.

The tables have overlapping information. It’s worth considering dropping either table 3 or table 4. It’s not clear to me why the numbers of infants receiving the vaccine in the windows isn’t the same in table 3 and 4.

- As detailed above, these tables provide complimentary information. We have revised the table captions to highlight and clarify the key differences. We have also substantially revised our results for clarity, and to better separate the information presented in these tables.

Why is the total of those in window 1 1212 and table 4 it’s 1203?

- In the previous draft, Table 3 showed that 1,212 unique participants had received more than 1 dose by the end of window 1 (including prior to window 1). Table 4 shows that 1,370 unique participants visited the clinic at least once during window 1, of which 1,203 received vaccinations (please note that the current draft includes minor updates to these specific numbers).

Discussion:

Page 8 Line 46 “Overall, we found that most infants received the full DTP3 series in a timely manner, indicating successful initiation into the EPI.” I think you need to say that this was only in study participants who returned for all 3 study visits.

- We have revised and expanded this sentence for clarity.

Reviewer 2: Ane Fisker, University of Southern Denmark

In their paper “Implementation and adherence of routine pertussis vaccination (DTP) in a low-resource urban birth cohort” the authors take advantage of the implementation of a longitudinal study

to monitor the pertussis incidence and assesses the DTP (Penta) vaccination coverage and timeliness.

While the close follow up certainly provides opportunities for a detailed description of the implementation programme, the extent to which the implementation of the vaccination programme has been affected by the study vaccines will have to be clarified.

The link between the study visits and vaccination services is not clear – are these always on the same date?

- Vaccinations were administered at three out of the six scheduled clinic visits. We have updated the introduction and methods to clarify this point.

Are the clinic visits scheduled at particular ages? I.e., the timing of the clinic visits every 2-3 weeks, how were these scheduled? From supplementary Fig S2, the density seems high at week 3, 6, 8, 10.5 and 12.5 determined.

- We have revised the methods to more clearly delineate the target and realized schedule.

Results seem to indicate that there are children “catching up” on their timelines, “Infants who missed a dose and later received a subsequent dose on (or near) schedule are also evident in Figures 1B-C. For example, 98 infants received their first DTP dose at age 10 weeks, while 160 infants received a second DTP dose at age 14-15 weeks (Figure 1C).”

- The examples quoted above describe infants who received late vaccinations. We note that potential catch-up is addressed at greater length in the discussion (para 5). A central concern is that the well-child vaccination schedule has no official catch-up mechanism for children who have aged beyond the scheduled 14 week vaccination. If doses are missing prior to that point, the odds that the child will receive a missed dose drop immediately, because there is no ‘scheduled’ contact with a child beyond that point until the 9 month measles dose. The system is simple, but not designed to be self-correcting. Our analysis highlights the consequences of this simplicity.

and P 10, I 16: “For example, the small change in unvaccinated infants attending window 2 (139, 10.7%) and window 3 (123, 11.4%) suggests that doses are not skipped randomly by study participants” I do not follow how this can be inferred from data on the individual vaccination – and not on the individual child.

- In this study we observe vaccinations of individual children, while the quoted text summarizes the behavior of the individuals in our cohort. Please note that Table 4 is stratified by age window, and the percentages shown are of visits within each age window. We note that, if a random sample of infants were vaccinated in each window, then the result would be a

decrease over time in the proportion of never-vaccinated infants across age windows. We do not see such a pattern, particularly between windows 2 and 3.

Half of the screened infants enter the study. It is reported that the main reasons for study exclusion were lack of consent (14.3%), low infant birth weight (7.9%), and community non-residence (5.9%) – but these numbers do not come close to the 50% excluded. Has something been forgotten?

- We have revised and added to this section to clarify the top five reasons, and referred to the original publication listing study enrollment details.

The numbers in text and figure S1 are not aligned. To make it clear what the reasons for non-participation are, I suggest to add these to Figure S1.

- Thank you, we have corrected the text. We have also added a reference to the original publication where full enrollment details are provided (Gill et al. 2016 Supplemental Table 3). For clarity of presentation, we have elected not to further detail the 14 reasons for exclusion in Figure S1.

“Prospective observation of a cohort can also reveal if community level DTP coverage differs substantially from regional coverage, and can help identify behavioral factors that affect the success of vaccine implementation programs, including vaccine hesitancy.” – But the authors do not address this. Could the clinical data be helpful in identifying if illness episodes made the mothers defer vaccinations?

- We do briefly address regional comparisons in the discussion (para 2), along with individual behavior (para 3). We agree that the impact of illness on vaccine uptake warrants further exploration. Unfortunately, this is outside the purview of the present text.

P 6. 44. “Data were de-identified prior to analysis: each mother-infant pair was assigned a random date offset of +/-3 days, and all study dates were adjusted by this offset.” Presumably “study dates” mean date of birth, date of visit and date of vaccination, but please clarify.

- Fixed.

The authors conclude their data can be used to identify spatial heterogeneity, but then goes on to compare their estimates with coverage at 12 months

- Our discussion does not identify or draw specific conclusions about spatial heterogeneities in this system. We do note that “spatial and socioeconomic heterogeneities in coverage are common [in LMICs]” (p2 L34-35), and that “our results suggest that spatial heterogeneity and delays in timely vaccination both warrant further attention in Zambia” (p9 L9-10). We agree (and specifically state) that our results are not directly comparable with up-to-date DTP3 coverage at 12 months (per WHO-UNICEF, p9 L1), and we further explore the question of make-up opportunities in this particular community (p9 L35-49).

Figure 2 – Better labels of the y-axis would make the figure easier to read.

- Fixed. The figure caption has also been revised to clarify the meaning of demographic covariates.

- TODO: re-upload Fig 2 & 3

Figure 3 – Last tables – I do not understand how 93,92 and 92 DTP1-2-3 children can be lost to FU at 4 weeks.

- We have revised the table captions to more clearly indicate the numbers shown here. These are individuals who attended a post-enrollment visit, but left the study prior to their first vaccination (see also Fig 1A). Please note that we identified and corrected a logic error in the “Attending” table (see, e.g., Week 18). We also identified 212 additional vaccination records (out of 3,567 total) that were erroneously omitted from the previous revision. Consequently, the tables embedded in Figure 3 have appreciably changed, though the previously described pattern remains largely unchanged.

P. 8, line 40: (save for?)

- Edited for clarity

VERSION 2 – REVIEW

REVIEWER	Mark McMillan 1.Vaccinology and Immunology Research Trials Unit, Women's and Children's Health Network, Adelaide, SA, Australia, 5006. 2.Robinson Research Institute and Adelaide Medical School, The University of Adelaide, Adelaide, SA, Australia, 5005.
REVIEW RETURNED	29-Sep-2020

GENERAL COMMENTS	Thank you for addressing the reviewer comments. Please see my minor suggestions below: Abstract: I think you are using DPT and DTP interchangeably. Stick with one for clarity. Results: Table 1, can you please put in what the abbreviation UTH stands for? Table 3, should the row for DTP1 equal the doses in Table 2? DTP 1 adds up to 1389.
--

REVIEWER	Ane Fisker Bandim Health Project and University of Southern Denmark
REVIEW RETURNED	04-Oct-2020

GENERAL COMMENTS	The authors have in their revised manuscript most of my prior doubts – but the identification of the 212 missed vaccines, raise a new concern. Since the authors presume full information on all vaccines (and therefore count children as unvaccinated for a particular dose if no vaccination is recorded), this assumption may not hold. Were these vaccines missed because they had been given at another health facility? If children who presented a vaccination card with additional, unregistered doses of vaccines at one of the clinic visits, had their vaccination status updated, providing information on the proportion who presented a card is important. (As children who did not present a vaccination card could not have their status updated).
--

	I have some further specific specific comments/suggestion for corrections below: Table 1: Born at UTH – what is UTH? OPV and BCG at birth – what does this mean – within how many days? Since the only factor associated with completing DTP3 is OPV at birth, and OPV at birth is lower than BCG at birth and lower than birth in HC, I wonder if reception thereof is an indicator for birth during a particular period? P7, l 9: Duration of study attendance is reported at 96 days – but the median as reported in Fig 1A seems to be 15 weeks (105 days) p 10 line 5: “The Kaplan-Meier survival analysis (assuming no vaccination in infants lost to follow-up)”, I think this is not correct, but you describe it correctly on Page 6 Line 33: The standard Kaplan-Meier method assumes that non-attending individuals (i.e., right-censored) experience events at the same rate as attending individuals. Table 3&4: Consider clarifying that the windows are distinct by 0-40, 41-69, 70-97, 98-128 Your suggestion of a serosurvey in 1-5 year olds (page 10, l. 53) will likely not say anything about late vaccinations. Typo in Supplementary figure. – Attending
--	--

VERSION 2 – AUTHOR RESPONSE

Reviewer: 1

Please see my minor suggestions below:

Abstract:

I think you are using DPT and DTP interchangeably. Stick with one for clarity.

Thank you for spotting this. Fixed.

Results:

Table 1, can you please put in what the abbreviation UTH stands for?

Fixed.

Table 3, should the row for DTP1 equal the doses in Table 2? DTP 1 adds up to 1389.

A single infant in our study received DTP1 at age 272 days, well outside the age windows shown in Table 3. The next latest vaccination age in our study was 128 days, and thus is included in both tables. We have clarified this point in the Table 3 caption.

Reviewer: 2

The authors have in their revised manuscript most of my prior doubts – but the identification of the 212 missed vaccines, raise a new concern. Since the authors presume full information on all vaccines (and therefore count children as unvaccinated for a particular dose if no vaccination is recorded), this assumption may not hold. Were these vaccines missed because they had been given at another health facility?

We apologize for the earlier omission, which arose from a programmatic error in our analysis code that came to light during the revision process. These 212 vaccinations have recorded dates that do not coincide with a recorded clinic visit for that individual (and were thus erroneously excluded in the first draft). We believe that two primary factors are responsible for the observed mismatches between clinic visit and vaccination dates. First, we expect that data entry errors were responsible for some date mismatches. In most cases, a clinic visit was recorded within several days of the vaccination, suggesting a transcription or clerical error. Second, no proximate clinic visit was apparent in a few cases, which could indicate either a lost clinic visit form or a vaccinee that bypassed the standard clinic visit.

We note that logistical issues arise in any field study, particularly in low-resource settings. Here we observe a 6% rate (212/3,567) of date mismatches. However, this issue does not suggest or indicate that unobserved vaccinations occurred at or near this rate. As previously noted and now highlighted in the manuscript, we expect that, due to study inducements and geographic and economic constraints of study participants, we successfully captured most clinic interactions during the study, and that non-clinic (for-cost) vaccinations of study infants were rare.

If children who presented a vaccination card with additional, unregistered doses of vaccines at one of the clinic visits, had their vaccination status updated, providing information on the proportion who presented a card is important. (As children who did not present a vaccination card could not have their status updated).

In the event that a child's under 5 card is lost, which certainly does occur, a new card was issued by the clinic. This would occur prior to giving a vaccination because the vaccine clinic will not give a dose unless it can document this on a card. It is possible a dose could have been given and documented but the card then lost before we saw that child next, in which case that the dose would be undocumented. We did not track the issuing of new cards and have no way to quantify this hypothetical. However, given that we saw the clients on 2 week intervals, the likelihood of this event seems very low, and any impact thereof would be minimal.

I have some further specific specific comments/suggestion for corrections below:

Table 1: Born at UTH – what is UTH?

Fixed

OPV and BCG at birth – what does this mean – within how many days?

Same day as birth, at the delivery site.

Since the only factor associated with completing DTP3 is OPV at birth, and OPV at birth is lower than BCG at birth and lower than birth in HC, I wonder if reception thereof is an indicator for birth during a particular period?

As with Table 1, the analysis presented in Fig 2 is primarily intended to address readers' concerns about potential stratification of our results by demographic covariates. Here we show 24 separate relative risk estimates with confidence intervals, and see no consistent patterns throughout. We note that a simple Bonferroni correction (e.g. $\alpha=0.05/24$) would not consider the association between DTP3 and OPV at birth statistically significant. We have revised the manuscript to highlight the marginal (and not statistically significant) nature of the reported associations.

P7, I 9: Duration of study attendance is reported at 96 days – but the median as reported in Fig 1A seems to be 15 weeks (105 days)

The median duration from study enrollment to completion was 96 days. We note that Fig 1A shows infant age. As the median enrollment age was 7 days, the age at which 50% enrollment was reached corresponds to slightly less than 15 weeks, as shown in Fig 1A.

p 10 line 5: “The Kaplan-Meier survival analysis (assuming no vaccination in infants lost to follow-up)”, I think this is not correct, but you describe it correctly on Page 6 Line 33: The standard Kaplan-Meier method assumes that non-attending individuals (i.e., right-censored) experience events at the same rate as attending individuals.

We have revised the results to clarify that we are presenting the modified, worst-case scenario that we discuss in the methods.

Table 3&4: Consider clarifying that the windows are distinct by 0-40, 41-69, 70-97, 98-128

Thank you, fixed.

Your suggestion of a serosurvey in 1-5 year olds (page 10, l. 53) will likely not say anything about late vaccinations.

We respectfully disagree. The WHO recommends a pertussis booster for children aged 1-6 years, noting that “the timing of this booster should also provide an opportunity for catch-up vaccination” [1], the schedule of which is determined by individual countries. Such a serosurvey could be compared to our results, to published national up-to-date estimates, and across geographical regions within countries to assess whether and where booster doses have increased rates of seroconversion. Such a survey could also provide strong evidence of the absence of late vaccinations, which could be particularly important in identifying vulnerable communities and appropriately targeting additional resources for booster / catch-up campaigns. We have revised the manuscript to highlight this point.

Typo in Supplementary figure. – Aattending

Fixed.

References

[1] World Health Organization. "Pertussis vaccines: WHO position paper." *Weekly Epidemiological Record= Relevé épidémiologique hebdomadaire* 85.40 (2010): 385-400.